# Experimental Study on Performance Enhancement of a Photovoltaic Module Incorporated with CPU Heat Pipe—A 5E Analysis

**DOI:** 10.3390/s22176367

**Published:** 2022-08-24

**Authors:** Seepana Praveenkumar, Aminjon Gulakhmadov, Ephraim Bonah Agyekum, Naseer T. Alwan, Vladimir Ivanovich Velkin, Parviz Sharipov, Murodbek Safaraliev, Xi Chen

**Affiliations:** 1Department of Nuclear and Renewable Energy, Ural Federal University Named after the First President of Russia Boris Yeltsin, 19 Mira Street, 620002 Ekaterinburg, Russia; 2Research Center for Ecology and Environment of Central Asia, Xinjiang Institute of Ecology and Geography, Chinese Academy of Sciences, Urumqi 830011, China; 3State Key Laboratory of Desert and Oasis Ecology, Xinjiang Institute of Ecology and Geography, Chinese Academy of Sciences, Urumqi 830011, China; 4Ministry of Energy and Water Resources of the Republic of Tajikistan, Dushanbe 734064, Tajikistan; 5Institute of Water Problems, Hydropower and Ecology of the National Academy of Sciences of Tajikistan, Dushanbe 72042, Tajikistan; 6Department of Automated Electrical Systems, Ural Federal University, 620002 Yekaterinburg, Russia

**Keywords:** photovoltaic, CPU fanless heat pipes, energy, exergy, embodied energy, LCE

## Abstract

As is already known, solar photovoltaic (PV) technology is a widely accepted technology for power generation worldwide. However, it is scientifically proven that its power output decreases with an increase in the temperature of the PV module. Such an important issue is controlled by adopting a number of cooling mechanisms for the PV module. The present experimental study assesses the effect of a fanless CPU heat pipe on the performance of a PV module. The experiment was conducted in June in real weather conditions in Yekaterinburg, Russian Federation. The comparative analysis of two PV panels (i.e., cooled, and uncooled) based on the electrical energy, exergy performance, economic, embodied energy and energy payback (5E) for the two systems is presented and discussed. The key results from the study are that the average temperature reduction from the cooling process is 6.72 °C. The average power for the cooled panel is 11.39 W against 9.73 W for the uncooled PV panel; this represents an increase of 1.66 W for the cooled module. Moreover, the average improvements in the electrical efficiency, and embodied energy recorded for a cooled PV panel 2.98%, and 438.52 kWh, respectively. Furthermore, the calculations of the levelized cost of energy (LCE) for the cooled PV panel indicate that it can range from 0.277–0.964 USD/kWh, while that for the uncooled PV panel also ranges from 0.205–0.698 USD/kWh based on the number of days of operation of the plant.

## 1. Introduction

Fossil fuels have been the major source of energy generation for a very long time now globally [1,2,3,4]. These fossil fuels have become an environmental concern, due to the negative effect they have on the environment [5,6,7,8]. Therefore, the demand for clean and renewable energy (RE) sources has in recent years increased around the world to help reduce the usage of fossil fuels [9,10,11]. Solar photovoltaic technology is one of the reachable, clean, and viable RE options that is broadly acceptable around the globe [12,13,14]. Solar PV generates electricity by converting solar energy directly into electrical energy, where solar radiation is available. Solar PV technology is noiseless during its operation and needs little maintenance [15,16]. Although solar PV technology is the most viable technology globally, the major disadvantage of PV is that as the cell temperature increases, its electrical efficiency also decreases [17,18,19,20]. Therefore, to maintain an appropriate electrical efficiency of the PV cell, it is necessary to provide appropriate cooling techniques to reduce its temperature, which will in the long run increase the lifespan of the PV module [21]. According to studies, heat-dissipating methods are categorized in two forms, i.e., active and passive cooling mechanisms [22,23,24]. Active cooling mechanisms use an external power source such as fans or heat pumps for heat dissipation, and they require extra space to mount the components [25,26]. In contrast, passive cooling mechanisms are relatively cheaper, more reliable, and more effective than active cooling mechanisms due to the use of heat sink technologies and ambient air for the extraction of heat from the PV module [26,27]. Consequently, to reduce the temperature and increase the lifetime of PV panels, researchers around the globe follow different modern technologies.

Researchers have proposed numerous methods for the cooling of solar PV panels that involve both active and passive cooling mechanisms. Perez et al. [28] presented experimental research using discontinuous fins to enhance the performance of a PV module. Their empirical study suggested that their proposed cooling method could lead to a reduction in the temperature of a PV panel by about 5.1 °C. Agyekum et al. [29] proposed a dual surface cooling method for a PV panel using a cotton wick. Preliminary results from their experimental work show that the average temperature drop between the cooled and reference module is 24 °C, and the overall improvement in electrical efficiency is 12%. Praveenkumar et al. [30] mounted aluminum sheets at the rear end of a PV panel. Their results suggested that the average temperature of the cooled PV panel was reduced by 10 °C, which translated into some 9.5% enhancement in the power output of the module. Similarly, Chen et al. [31] also used fins to improve the efficiency of PV panels. Yousuf et al. [32] conducted an experimental investigation into PV, PV/ PCM, and PV PCM/AF modules. According to their study, the integration of a PV module with a PCM/AF resulted in a temperature decrease of about 37%, and an electrical efficiency enhancement of 14%. In other studies, Bayrak et al. [33] examined three different situations to cool the PV module between PCM, TEM, and Al fins. Based on their experimental work, the integration of fins to the PV system generated the highest power output while the PV system with PCM and TEM recorded the lowest output. Mays et al. [34] studied the performance of a PV using an aluminum finned plate. Their study shows that the average temperature reduction of a PV cooled panel integrated with their proposed cooling method is 7 °C, which enhanced the power output by 1.86 W. Anna et al. [35] conducted a study on the performance of PV panels using four different PCMs. Their study concludes that the addition of a PCM layer on the rear side of a PV panel could reduce the temperature of the module and increase its electrical efficiency. Similarly, Sajjad et al. [36] used air from an air conditioner to enhance the performance of a PV panel. The study showed that the adopted method could increase the PV cell’s electrical efficiency by 6%; similar work has been done by Bashir et al. [37] in Pakistan. Dida et al. [38] studied the effect of a PV module using a burlap cloth. The results showed that the panel’s temperature decreased by 20 °C, leading to an increase in efficiency of 15%. Deokar et al. [39] proposed a new active cooling technique using mild steelchips. A 16 °C temperature was occasioned as a result of the integration of their proposed cooling method leading to a 12.3% increase in the efficiency of the module. Stefan [40] also proposed the use of a thin film of water through the front surface of the PV panel. Their results concluded that the average efficiency of the panel increased by 10%, and the average temperature reduced by 22 °C. You et al. [41] developed an indoor organic PV (OPV) by using a TiO_2_ layer and metal oxide metal. According to the researchers, the power conversion efficiency of OPV increasd by 8.8%. Saeed et al. [42,43] also discussed the recent developments in the indoor organic photovoltaic cells industry. Furthermore, authors such as Al-Amri [44], Elnozahy et al. [45], Chandreskar et al. [46], Amr et al. [47], Kidegho et al. [48], Tsankas et al. [49], and Hernández-Callejo [50] assessed the potential of using various cooling methods to cool PV modules to help enhance their performance. It is evident from the literature reviewed above that a number of studies have been conducted to assess the potential of several cooling methods for the PV module. Therefore, this study also examined the performance of a PV panel using fanless heat pipe CPU heat sinks to cool the temperature of the PV module. Moreover, this study was performed in real weather conditions in Yekaterinburg, Russian Federation located at latitude: 56.841 °N, longitude: 60.64 °E. The outdoor experiment comprised a modified PV panel with CPU heat pipes at its rear side (i.e., a cooled panel), and a PV panel without any modifications (i.e., an uncooled panel). Parameters such as temperature distribution, and energy, exergy, embodied energy, economic and energy payback (5E) analysis for the two modules are presented and discussed in the paper.

## 2. Materials and Methods

This section presents the working principle of a fanless heat pipe CPU sink, and the construction of an experimental test rig. It also contains the mathematical relations used for the calculation of the various parameters. The two modules used for the experiment (i.e., cooled and uncooled) have a capacity of 30 W each with a length of 950 mm and a width of 450 mm.

### 2.1. Heat Pipe Theory and Operation

The most essential technology that helps in the cooling of electrical equipment is air conditioning [51]. There were three main ways to cool electronic equipment in the past: (1) passive air cooling, which dissipates heat by forcing air to flow using fans; (2) forced air cooling, which dissipates heat by forcing coolants such as water to pass [52]; and (3) forced liquid cooling, which dissipates heat by forcing coolants such as water to pass [52]. Forced convection, which involved directly connecting a fan to a heat sink, was the traditional method for dissipating heat from desktop computers. Heat sinks with plate fins are particularly useful in cooling electronic equipment because of their advantages, such as simple machining, simple structure, and cheaper cost [53]. Heat flux for the CPU has increased dramatically as a result of the reduced CPU size and increased power found in modern computers [54]. Limits on the size of heat sinks and fans, as well as the noise level associated with increasing fan speed, have been enforced. As a result, there has been an increasing demand for better cooling solutions that are compatible with today’s CPU requirements. Two-phase cooling systems, such as the heat pipe and the thermosyphon, have emerged as viable heat transfer devices as alternatives to traditional heat sinks, with effective thermal conductivity over 200 times that of copper [55].

To overcome pressure drops within the heat pipe, the highest capillary pressure must be higher than the sum of all the pressure drops inside the heat pipe; hence, the primary condition for heat pipe operation is as follows:(1)$$\Delta P_{c}\geq \Delta P_{1}+\Delta P_{v}+\Delta P_{g}$$
where $\Delta P_{c}$ denotes the maximal capillary force within the wick structure, and $\Delta P_{1}$ denotes the pressure drop required to return the liquid from the condenser to the evaporation section. $\Delta P_{v}$ is the pressure drop required to transfer vapour from the evaporation to the condenser section, and $\Delta P_{g}$ is the pressure drop induced by a difference in gravitational potential energy (which can be positive, negative, or zero depending on the heat pipe orientation and direction) R With reference to Figure 1 [56], the basic processes of heat pipe operation are as follows:
The evaporation of the working fluid is enabled by the heat added at the evaporator portion by conduction through the wall of the heat pipe.Movement of vapor from the evaporator section to the condenser section occurs; this is influenced by the vapour pressure drop occasioned by the working fluid evaporation.In the condenser part, the vapour condenses, releasing its latent heat of evaporation.The liquid moves back to the evaporator section from the condenser section through the wick using capillary force and liquid pressure drop.

The liquid pressure, vapour pressure, and capillary pressure drops can be calculated from Equations (2)–(5) [57].
(2)$$\Delta P_{1}=\frac{\mu_{1}L_{eff}Q}{\rho_{1}KA_{w}h_{fg}}$$
(3)$$\Delta P_{v}=\frac{16\mu_{v}L_{eff}Q}{2{\left(\frac{D_{v}}{2}\right)}^{2}A_{v}P_{v}h_{fg}}$$
(4)$$\Delta P_{c}=\frac{2\sigma_{1}}{r_{eff}}$$
(5)$$Q_{max}=\left(\frac{\rho_{1}\sigma_{1}h_{fg}}{\mu_{1}}\right)\left(\frac{A_{w}K}{L_{eff}}\right)\left(\frac{2}{r_{eff}}-\frac{\rho_{1}gL_{eff}Sin\emptyset }{\sigma_{1}}\right)$$

In the horizontal direction if φ = 0 then the Equation (5) will be modified to Equation (6) as shown below
(6)$$Q_{max}=\left(\frac{\rho_{1}\sigma_{1}h_{fg}}{\mu_{1}}\right)\left(\frac{A_{w}K}{L_{eff}}\right)\left(\frac{2}{r_{eff}}\right)$$

Here, *μ*_l_, *μ_v_*_,_ signify liquid and vapor viscosity, $\rho_{1}$ & $\rho_{v}$ are liquid and vapor density, $A_{w}$ & $A_{v}$ are wick and vapor cross-sectional areas, respectively. Furthermore, g, $D_{v}$, *K,* $L_{eff}$, $h_{fg}$, $\sigma_{1}$, ∅, $r_{eff}$ signify gravity, vapor distance, wick permeability, effective length, heat of vaporization of liquid, surface tension, angle to the pipe in the horizontal direction, and effective radius of the pores of the wick, respectively [57].

### 2.2. Construction of Experimental Test-Rig

The construction of the experimental test-rig is shown in Figure 2. It includes a 60 mm × 40 mm × 10 mm Al sheet on which the fanless heat pipe sinks supplied by the Semoic company, China [58] were mounted for the cooled PV panel. The HY-170 thermal paste (grease) is applied between the back of the module and the aluminum sheet to increase the thermal conductivity between them [59]. A universal silicone gel was also used between the PV panels and Al sheet to hold them firmly [59]. A total of four fanless heat pipe sinks were mounted on the top of the Al sheet with the help of a 450 mm × 30 mm connecting rod at the back of a PV as shown in Figure 2. In addition, three sets of K-type thermocouples with a temperature range between −270 °C and 1260 °C with a resolution of 0.75% were also used to measure the temperature of the panels with the help of a 4-Channel SD Logger 88598 (World wise testing service Co. Ltd., Taipei, Taiwan) [60]. The thermocouples were manufactured by the REOTEMP instrument cooperation [61]. In order to perform further experimental works, a rectangular basin with a length and width of 950 mm × 450 mm × 350 mm was used to host the water and the integrated fanless heat pipe. In addition, a TM-207 solar pyranometer (Tenmars Electonics co. Ltd., Taipei, Taiwan) [62] was used to measure the solar radiation on the day of the experiment. A clamp meter (RS components, UK) [63] was employed to record the voltage and current of the two PV panels and a digital anemometer was employed to measure the wind speed during the experiment. The specifications of the four fanless heat pipes are presented in Table 1. The image of one of the fanless heat pipe sinks that was used for the present study is shown in Figure 3.

A picture of the set-up for the experiment and the schematic representation of the experiment are presented in Figure 4.

### 2.3. Energy Analysis

According to the first law of thermodynamics, the efficiency of solar PV panels is affected by the ambient temperature as well as the module temperature. Therefore, the energy efficiency of a PV panel is defined as the ratio of power output to the power input of a PV panel as shown below [30,59,65,66].
(7)$$P_{out}=I_{mp}\times V_{mp}$$
where $I_{mp}$ and $V_{mp}$ represent the current ampere and voltage, respectively.
(8)$$P_{in}=G\times A$$
where *G* is the global solar irradiation (W/m^2^), and A is the module area (m^2^); the area of the module is 0.4275 m^2^ used in the study [65].

Therefore, the energy efficiency ($\eta_{energy}$) of the PV module is calculated as follows [30,59,66];
(9)$$\eta_{energy}=\frac{P_{out}}{P_{in}}=\frac{I_{mp}\times V_{mp}}{G\times A}=\frac{V_{oc}\times I_{sc}\times FF}{GA}\;\;$$

The $V_{oc}$ is known as opencircuit voltage, $I_{sc}$ and *FF* are the short-circuit voltage and fill factor, respectively.

An increase in the PV cell temperature decreases both the open-circuit voltage and the *FF* while the short-circuit current increases but only slightly. Therefore, the net effect will result in a linear relation, as shown in Equation (10) [66].
(10)$$\eta_{c}=\eta_{T_{ref}}[1-\beta_{ref}\left(T_{c}-T_{ref}\right)+\gamma log_{10}I\;\left(t\right)$$

The solar coefficient is usually taken as zero or neglected and as a result Equation (10) reduces to Equation (11).
(11)$$\eta_{c}=\eta_{T_{ref}}[1-\beta_{ref}\left(T_{c}-T_{ref}\right)$$

Finally, the improvement in cooled PV can be computed by using Equation (12) [67].
(12)$$\eta_{improvement\;}\left(\%\right)=\left(\frac{\eta_{cooled,\;PV}}{\eta_{uncoold,PV}}-1\right)\times 100$$
where, $\eta_{T_{ref}}$ is the efficiency at STC taken as 15% in the current study $\beta_{ref}$ presents the temperature coefficient, and the value is 0.004/K, γ is the solar radiation coefficient, and the value is 0.12 [66,68]. Using PV_syst_ software (developed by PV SOL, Satigny, Switzerland), the effects of cell temperature on the characteristics of a 30 W generic poly PV module are shown in Figure 5 and Figure 6.

### 2.4. Exergy Analysis

According to the second law of thermodynamics, the exergy balance for a PV syst can be represented mathematically as presented in Equation (13) [32,69,70].
(13)$${\dot{\Sigma }}_{x_{out}}={\dot{\Sigma }}_{x_{in}}$$
where ${\dot{\Sigma }}_{x_{out}}$ is the exergy outlet rate, and ${\dot{\Sigma }}_{x_{in}}$ is the exergy inlet rate.

The exergy inlet from the sun can be computed using Equation (14) [71]
(14)$${\dot{\Sigma }}_{x_{in}}=\;{\dot{\Sigma }}_{x_{sun}}=\left(1-\frac{T_{amb}}{T_{Sun}}\right)GA$$
where $T_{amb}$ is the ambient temperature on the day of the experiment, which was measured using the GM 1362-EN-01 thermometer; $T_{Sun}$ is assumed to be 5770 K for the study, G is the solar insulation (W/m^2^), A is the area of the module [72].

The output exergy is defined as [73]:(15)$${\dot{\Sigma }}_{x_{out}}=I_{mp}\times V_{mp}-\left[\left(1-\frac{T_{amb}}{T_{cell}}\;\;\right)h{\;}_{c}\;A\;(T_{cell}\;-T_{amb)}\right]$$

Finally, the exergy efficiency for the PV system is defined as follows [74]:(16)$$\eta_{system}=\frac{{\dot{\Sigma }}_{x_{out}}}{{\dot{\Sigma }}_{x_{in}}}=\frac{I_{mp}\times V_{mp}-\left[\left(1-\frac{T_{amb}}{T_{cell}}\;\;\right)h{\;}_{c}\;A\;(T_{cell}\;-T_{amb)}\right]}{\left(1-\frac{T_{amb}}{T_{Sun}}\right)GA\;\;\;\;\;}$$
where $T_{cell}$ (K) is the surface temperature of the PV module, $h{\;}_{c}$ is the convective heat transfer coefficient (W/m^2^-K) and it is calculated by using the wind speed given in Equation (17) [75].
(17)$$h{\;}_{c}=5.7+3.8v$$

### 2.5. Economic Analysis

The economics of the cooled and uncooled modules were done using the levelized cost of energy (LCE) metric. According to other studies, LCE is a fundamental metric that is used in the calculation of the cost of renewable and non-renewable energy projects. The main objective of LCE in the current work is that it will determine whether to move further with the project or as a means to compare different energy-producing projects. Mathematically, LCE is expressed in Equations (18)–(23) [76,77,78].
(18)$$\mathrm{LCE}=\frac{LC_{O\&m}+LC_{fuel}+LC_{inv}}{E_{annual}}$$
(19)$$LC_{inv}=\mathrm{CRF}\times C_{inv}$$
(20)$$CRF=\frac{i_{eff}\times {\left(1+i_{eff}\right)}^{n}}{\left({\left(1+i_{eff}\right)}^{n}\right)-1}$$
(21)$$LC_{O\&M}=C_{O\&M}\times \mathrm{CELF}$$
(22)$$\mathrm{CELF}=\left(\frac{1+r_{n}}{1+i_{eff}}\times \frac{1-K_{o\&m}^{n}}{1-K_{O\&M}}\right)\times \mathrm{CRF}$$
(23)$$K_{O\&M}=\frac{1+r_{n}}{1+i_{eff}}$$

### 2.6. Energy Payback Time (EPBT)

The EPBT can be explained as the required time within which the energy savings recompense the invested energy. The invested energy in this case refers to the embodied energy $E_{in}$ which can be defined as the entire spent energy in the course of the manufacturing of a system over the whole lifecycle. One of the main indicators in identifying the sustainability of a certain RE power plant over other technologies is the EPBT. It can be estimated using Equation (24) [32].
(24)$${\left(\mathrm{EPBT}\right)}_{en}=\frac{E_{in}}{En_{out}}$$

### 2.7. Uncertainty Analysis and Experiment Measurement Assessments

In this section, the uncertainties associated with the experimental work are estimated. The following devices were used to record the various data from the experimental work: a pyranometer, a thermocouple, a clamp meter, a thermometer, and a digital anemometer. The standard uncertainty $F_{z\;}$ can be estimated using Equation (25) [79,80].

Where, $Y_{z}$ is the accuracy of the devices used in the experiment and that can be obtained from the manufacturer’s data sheet. Therefore, the uncertainty *X(b)* can be achieved using Equation (26). Table 2 represents the range, accuracy, and uncertainty of the devices. The total uncertainty error achieved for the present experiment is 3.97%.
(25)$$F_{z\;}=\frac{Y_{z}}{\sqrt{3}}$$
(26)$$\begin{array}{cc} X\left(b\right)=Square\;root & of\;[{\left(Uncertainty\;of\;Pyranometer\right)}^{2}\\ & +\;\;{\left(Uncertainty\;of\;\mathrm{thermocouple}\right)}^{2}\\ & +\;\;\;\;{\left(Uncertainty\;of\;clamp\;meter\right)}^{2}\\ & +\;\;\;{\left(Uncertainty\;\;of\;thermometer\right)}^{2}\\ & +\;\;\;\;(Uncertainty\;of\;digital\;\;anemometer^{2})] \end{array}$$

## 3. Results and Discussion

In this section, the obtained results from the experiment such as weather characteristics, thermal analysis, electrical improvement, economic, and energy payback time analysis are presented and discussed.

### 3.1. Weather Characteristics of the Experimental Period

The details of the weather on the day of the experiment are presented in this section. The solar radiation, ambient temperature, humidity, and wind speed are recorded from morning 08:30 h to 16:30 h within 30 min intervals of time, as shown in Figure 7 and Figure 8. The average solar radiation, ambient temperature, and relative humidity are 999.25 W/m^2^ 27.38 °C, and 38.54%, respectively. The wind speed ranged between 4.0–6 m/s. The highest solar radiation was recorded at noon, which was 1379 W/m^2^.

### 3.2. Performance of FanLess CPU Heat Sink on the PV Panels

In this section, we discussed the thermal and electrical performance of the two PV panels.

#### 3.2.1. Effect of Temperature on PV Panels

The PV temperature is a significant parameter for PV panels; it plays a fundamental role in identifying the system’s performance. This study used two panels, one modified PV panel with a fanless heat sink and another PV panel for comparison. Three K-type thermocouples were used at different locations on each PV panel, giving readings every 30 min. The temperature profile of the two tested PV panels is shown in Figure 9. The reduction in temperature is also presented in Figure 9. From Figure 9, as the day starts, the temperature of the cooled and uncooled PV panels increased until it hit its peak value at 13:30 h. The maximum temperature for the cooled panel during the experiment is found to be 50.26 °C against 60.19 °C for the uncooled panel. The average temperature reduction achieved between the cooled and referenced modules at the end of the experiment was found to be 6.72 °C. This reduction is relatively significant, especially when this process does not require electric power to cool the PV module; it also requires very little water for the cooling process and therefore can be employed in areas with water scarcity.

The thermal image profiles for both PV panels were recorded around 11:30 am on the day of the experiment. The findings from the thermographic images are presented in Figure 10 and Figure 11. The results from the thermal image show that the temperature of the cooled module ranges between 20–30 °C, that of the referenced module ranges between 31–35 °C. The positive impact of the cooling approach adopted is clearly shown in the thermal images. This confirms the earlier results obtained through the use of the thermocouples.

#### 3.2.2. Electrical Performance of a PV Panel

The voltage and current results for the cooled and uncooled panel are illustrated in Figure 12. According to the data, the maximum voltage achieved for the modified PV panel is 18.99 V, recorded at 12:00 h. Whereas, for the uncooled solar PV panel, its maximum voltage was recorded around 11:30 h. However, there is a voltage drop for the un-cooled panel as a result of its relatively high temperature. The average voltage for the cooled and the uncooled PV panels during the experiment was 18.36 V and 17.01 V. It shows that the temperature negatively influences the voltage of the uncooled panel. Figure 13 explains the power output of the tested system during the day of the experiment. The total power output of the PV panels increases as the day progresses; the power output from both panels increases with time thanks to solar insulation until midday. The trend of the power output, however, starts decreasing after midday due to the increasing panel temperatures and reduction in the intensity of solar radiation. The results revealed that the average power of the cooled PV panel is 11.39 W, as against 9.73 W for the module.

### 3.3. Electrical Efficiency

The electrical efficiency of the conventional PV panels (uncooled) over modified PV panels (cooled panel) is presented in Figure 14. According to the obtained results, the electrical efficiency for both PV panels experiences a downward trend from the start of the experiment until after 13:30 h due to the increasing temperature of the PV modules during that period. The trend, however, reversed after 13:30 h when both the ambient temperature and the PV module’s temperature began to decrease. The average electrical efficiency recorded for the period of the experiment for the cooled PV panel is 14.05%, against 13.65% for an uncooled PV Panel. The average improvement in electrical efficiency is about 2.98%. The present proposed approach is compared with other published literature as shown in Table 3. It is clear from the literature presented in Table 2 that the current study is either better in terms of results or equal to other forms of cooling methods proposed by other studies.

### 3.4. Exergy Efficiency

The exergy efficiency results achieved from the cooled and uncooled PV panels are presented in Figure 15. A PV system’s exergy efficiency is adversely affected by its power output. It can therefore be seen from the results that the profile for the exergy efficiency for the two modules follows the same trend as that of the electrical efficiency. Due to the increase in PV module temperature, the exergy efficiencies declined from the beginning of the experiment until after 13:30 pm, before it started rising again. The exergy of the cooled remained higher at all times during the experiment which suggests that the proposed cooling method is able to keep the temperature of the PV module under control. Consequently, the average exergy efficiencies for cooled PV and uncooled panels are 7.88% and 4.54%, respectively. From the results, it is evident that the modified PV panel recorded relatively high exergy efficiency.

### 3.5. Economic Analysis

Table 4 depicts the cost of the various items used for the construction of the PV panels. Table 5 depicts the estimated LCE of a PV plant. The experiment was performed in the Russian Federation, where there are poor climatic conditions. Therefore, we assumed two scenarios to calculate the LCE. In the first scenario, the effective period for Russian conditions starts from May to the middle of August, which is about 105 days a year; this is the period with the best weather conditions in the area where the experiment is conducted. The second scenario i.e., 365 days, assumes that the entire year experienced good weather conditions for the generation of electrical energy from the PV module. Furthermore, it is also assumed for the purposes of this estimation that the PV modules worked effectively for a period of 10 h daily. Through the experiment, it was found that the power generated by the cooled PV panel is 11.39 W against 9.73 W by the uncooled panel.

#### 3.5.1. Scenario 1 for 105 Days

For the purposes of this calculation, it is also assumed in this study that equal power is generated throughout the year. This means a total electricity of 11.95 kWh and 10.216 kWh would be generated by the cooled and uncooled modules, respectively, for scenario 1. Using the data provided in Table 5 and Table 6, the LCE of the cooled and uncooled modules are estimated to be 0.96 USD/kWh and 0.61 USD/kWh.

#### 3.5.2. Scenario 2 for 365 Days

The power that would be generated for the entire year (i.e., 365 days) for the cooled PV panel is 41.5735 kWh, and for the uncooled panel is 35.5145 kWh. The results suggest that for the 365 days, the LCE of the cooled PV panel is 0.277 USD/kWh and 0.206 USD/kWh for the uncooled panel.

### 3.6. Energy Payback Time

The embodied energy computations achieved for the cooled and uncooled PV panels are presented in Table 6. Due to the extra materials used in the cooling of the cooled PV, its embodied energy is found to be greater than that of the uncooled PV. The embodied energy for the cooled and uncooled modules were found to be 438.53 kWh and 427.50 kWh, respectively. The estimated result show that the EPBT (energy base) for the cooled PV panel is 10.54 Year against 12.16 Year for the uncooled PV panel. Based on the EPBT results, it can be seen that the embodied energy for the extra materials used for the cooling process affected the EPBT years for the cooled module. It therefore suggests that appropriate mechanisms have to be put in place to reduce the embodied energies of the various materials used for the construction. Manufacturers need to produce materials with lower embodied energy.

## 4. Conclusions

In this study, the fanless heat pipe sink was employed for thermal management of a PV system. The main objective of this study was to evaluate the effectiveness of using a fanless heat pipe sink to cool and enhance the performance of a PV module. The electrical energy, exergy, economics, embodied energy and the energy paybacks of two PV modules (i.e., cooled and uncooled) were evaluated. From the present research, the following significant conclusions have been drawn from the experiment:(1)The average temperature of a cooled PV panel for the experimental period is 40.76 °C, while that of the referenced PV panel is 47.49 °C. Thus, a temperature reduction of 6.73 °C was obtained as a result of the cooling of the PV module.(2)The average voltage and current for the cooled PV panel during the experimental day are 18.36 V and 0.619 A, while those for the uncooled PV panel are 17.01 V and 0.57 A, respectively.(3)The average power output recorded for the cooled PV panel is 11.39 W and 9.73 W for the reference panel. The overall improvement in the production of power is 1.66 W.(4)The average exergy efficiency difference between for a cooled and an uncooled PV panel is 3.34%.(5)results that for the first scenario, i.e., 105 days, for the cooled PV panel would be 0.96 USD/kWh against 0.61 USD/kWh for the uncooled module. Furthermore, in the case of the second scenario, i.e., 365 days, the LCE recorded for the cooled PV panel equals 0.277 USD/kWh compared to 0.206 USD/kWh for the reference panel.(6)The cooled panel was found to have an embodied energy of 438.525 kWh while that of the uncooled model was 427.5 kWh, which translates into an EPBT of 10.54 years and 12.16 years for the cooled and uncooled PV modules, respectively.

The proposed cooling mechanism for reducing the PV panel temperature proved to be adequate. The LCE acquired for the present experiment is a bit higher for the cooled panel because of the extra cost associated with the materials used for its construction. Therefore, we recommend that manufacturers produce materials for the market with lower embodied energy due to the development of advanced technologies.

## Figures and Tables

**Figure 1 sensors-22-06367-f001:**
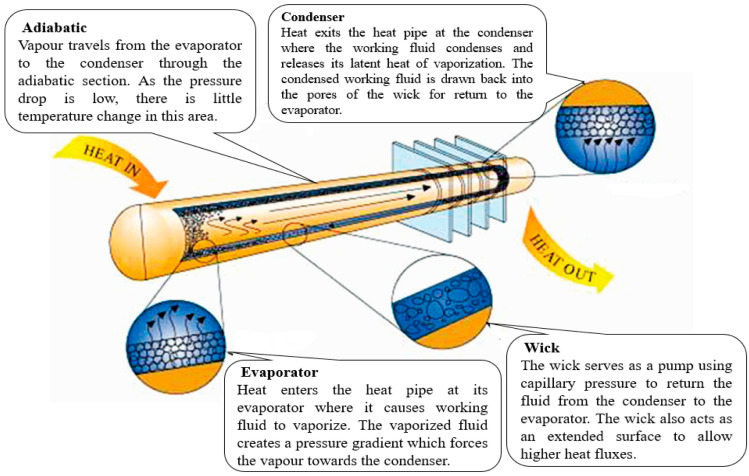
Heat pipe operation [56].

**Figure 2 sensors-22-06367-f002:**
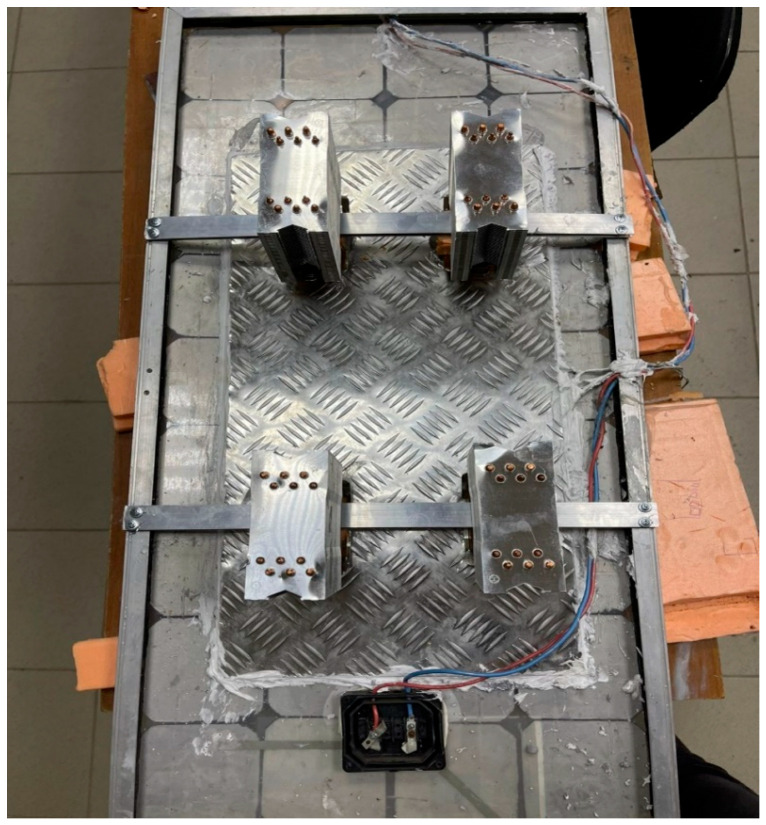
Modified PV panel with integrated fanless heat pipe sink.

**Figure 3 sensors-22-06367-f003:**
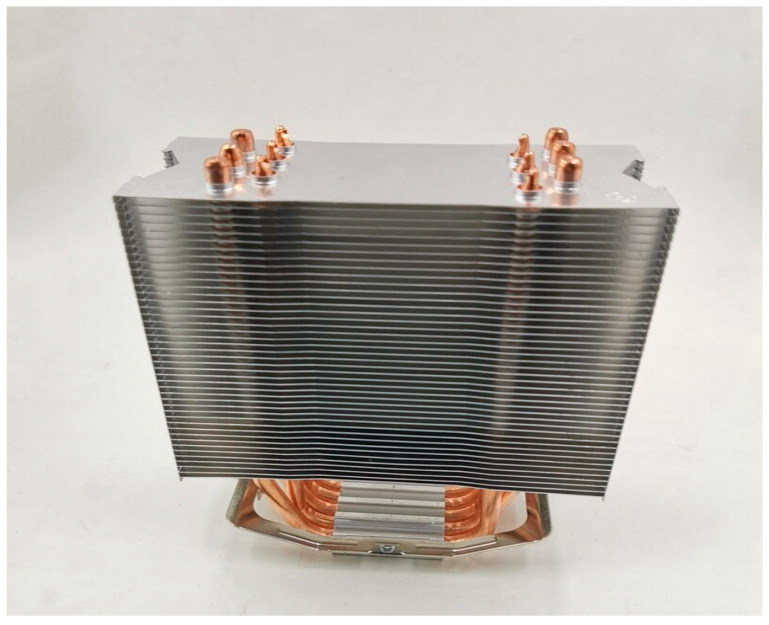
Fanless heat pipe sink.

**Figure 4 sensors-22-06367-f004:**
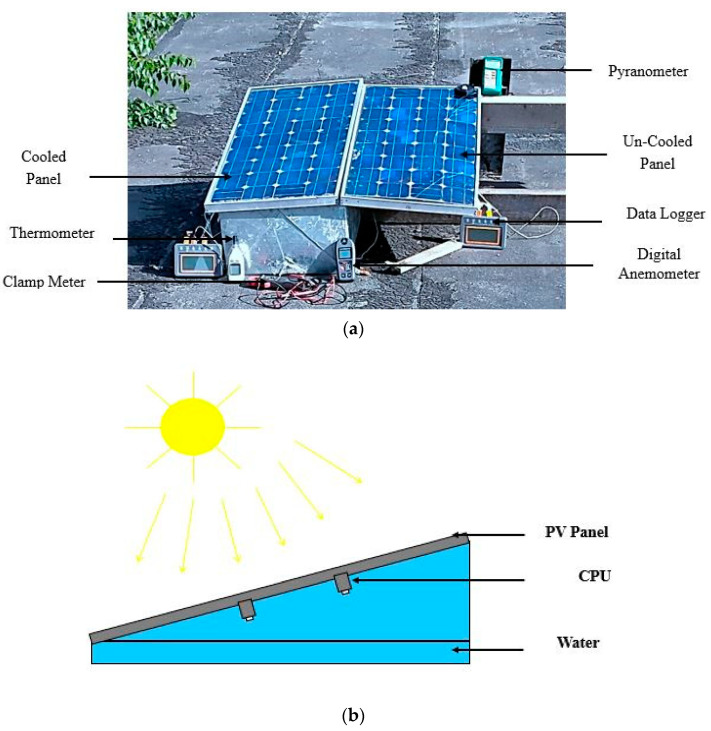
(**a**) Experimental setup (**b**) Schematic diagram of experimental test rig.

**Figure 5 sensors-22-06367-f005:**
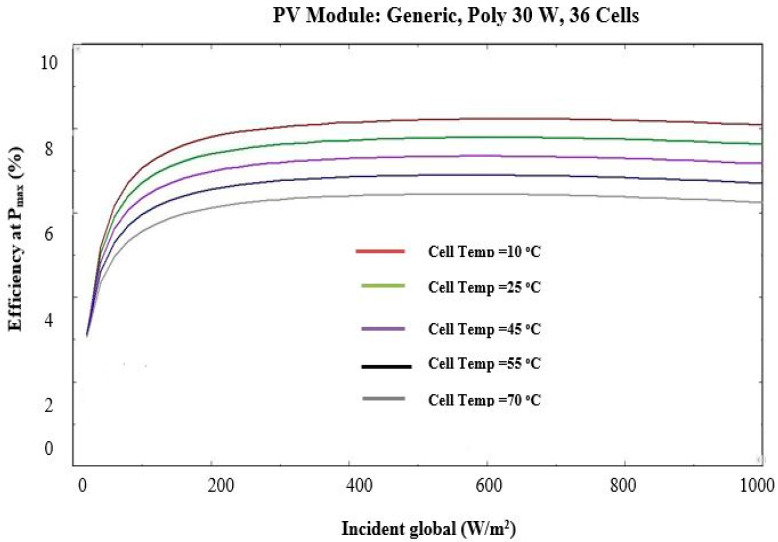
Effect of temperature on efficiency and global solar radiation (obtained from PVsyst software).

**Figure 6 sensors-22-06367-f006:**
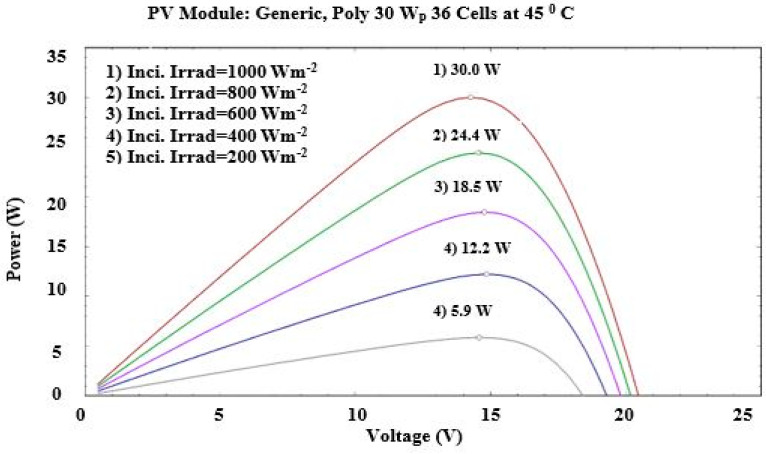
Effect of incident radiation on the P-V plot at 45 °C cell temperature (obtained from PVsyst software).

**Figure 7 sensors-22-06367-f007:**
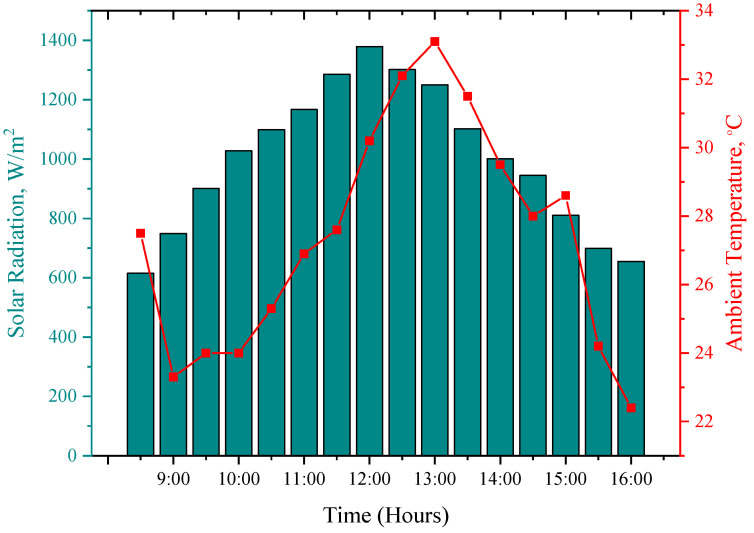
Weather characteristics for the period of the experiment (i.e., solar radiation and ambient temperature).

**Figure 8 sensors-22-06367-f008:**
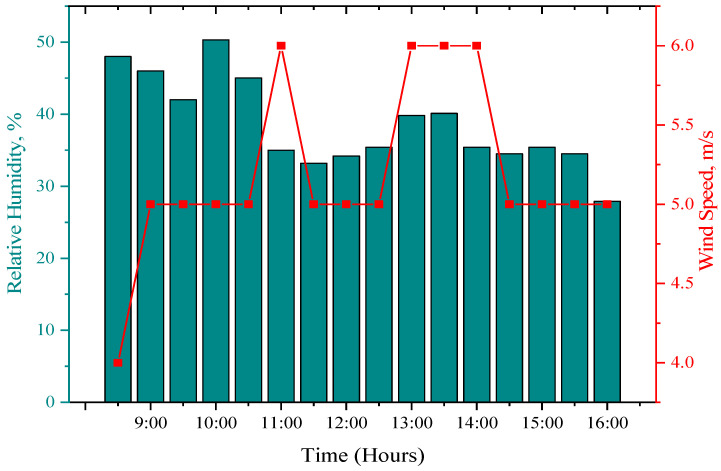
Weather characteristics for the period of the experiment day in relative humidity and wind speed.

**Figure 9 sensors-22-06367-f009:**
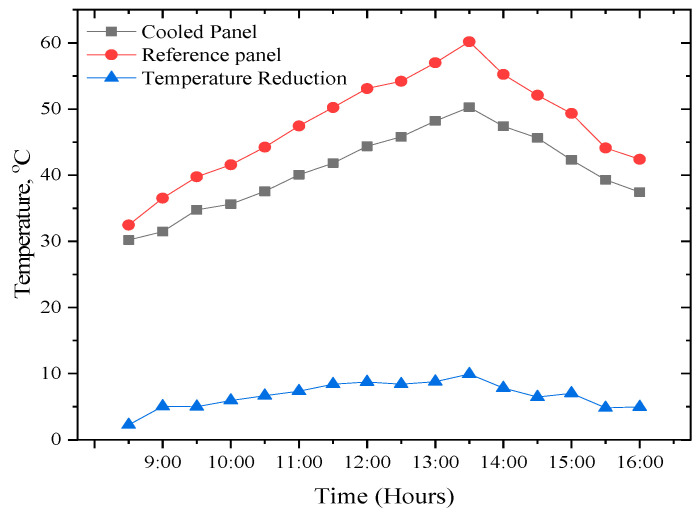
Time dependence of temperature of the two PV panels.

**Figure 10 sensors-22-06367-f010:**
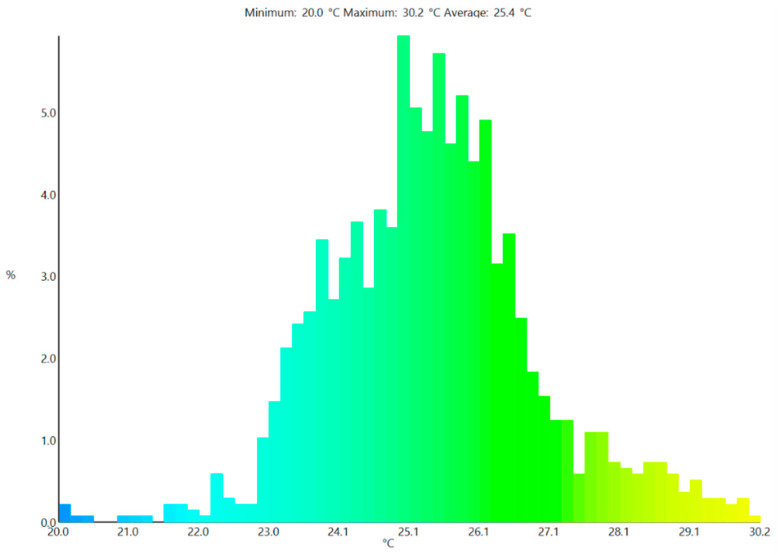
Thermal image of a cooled PV panel.

**Figure 11 sensors-22-06367-f011:**
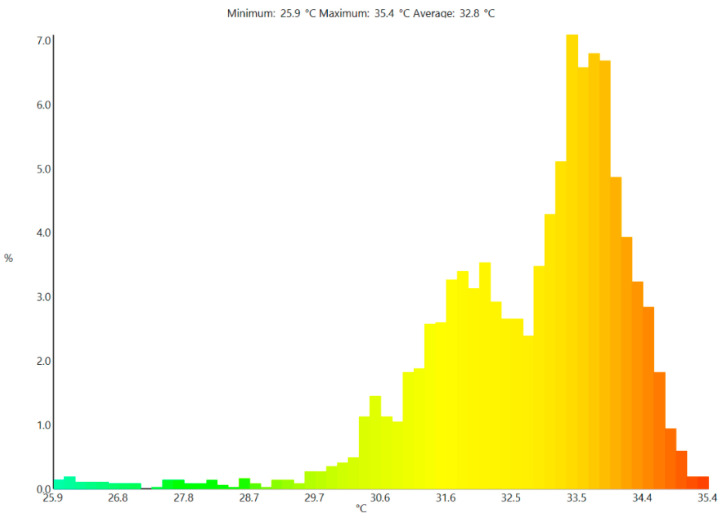
Thermal image of an uncooled PV panel.

**Figure 12 sensors-22-06367-f012:**
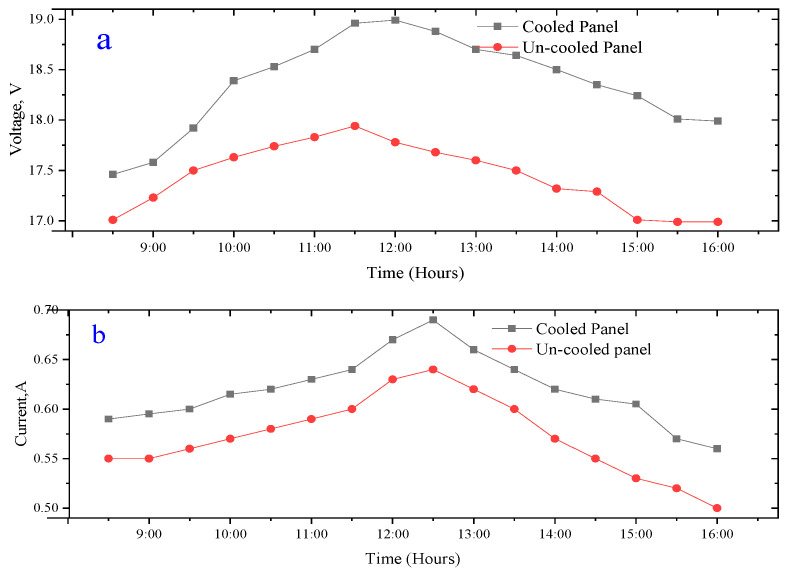
Time dependence of (**a**) Voltage and (**b**) Current of both PV panels.

**Figure 13 sensors-22-06367-f013:**
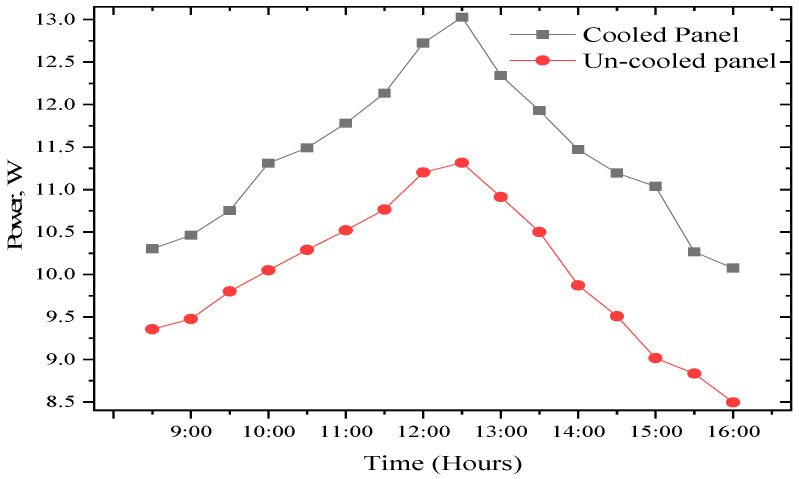
Temperature dependence power output of both PV panels.

**Figure 14 sensors-22-06367-f014:**
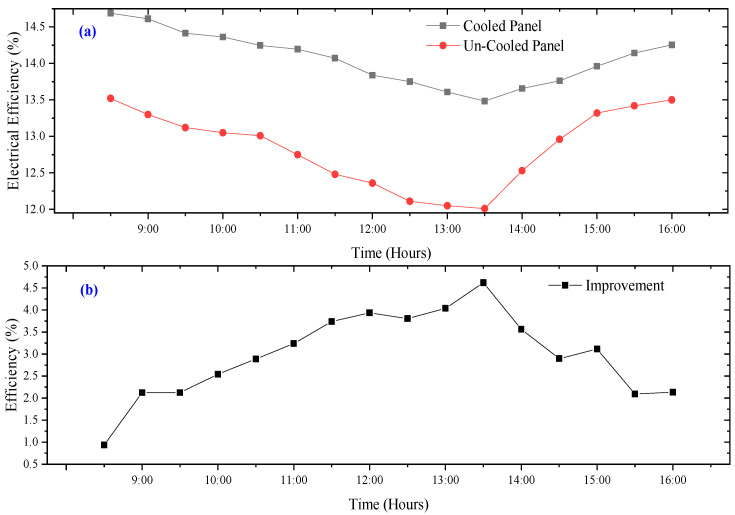
Time dependence (**a**) electrical efficiency (**b**) improvement in efficiency.

**Figure 15 sensors-22-06367-f015:**
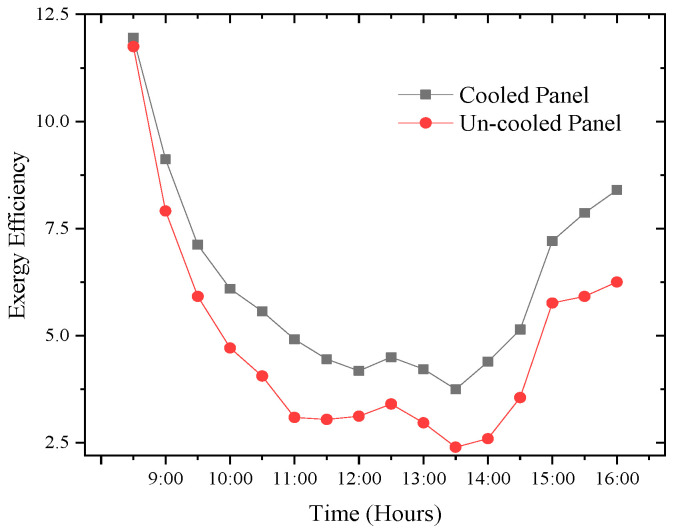
Time-dependent exergy efficiency.

**Table 1 sensors-22-06367-t001:** Specifications for the fanless heat pipe sink [58,64].

Specifications	Value
Origin	China
Model Name	249761
Manufacturer	Seomic
Size (L × W × H), mm	98 × 95 × 135
Cooling Type	Air Cooler
Material	Metal
Use	CPU enhance cooling
Color	Silver
Measurement error	+/−1–3 cm

**Table 2 sensors-22-06367-t002:** Uncertainties of Measuring Instruments [67,81].

S. No	Instrument	Units	Range	Accuracy (%)	Uncertainty (%)
1	TM-207 Pyranometer	W/m^2^	0–2000	±5	2.886
2	K-Thermocouple	°C	–270 to 1260	±0.75	0.433
3	Clamp Meter		-	±3%	1.732
4	Thermometer	°C	−30 to 70	±2	1.15
5	Digital anemometer	m/s	0–30	±3	1.732

**Table 3 sensors-22-06367-t003:** Comparison of other published works.

Reference	Type of Cooling	Proposed Mechanism	Key Results
[82]	Active Cooling	Heat exchanger	Output power is increased by 2.94 W. Improvement in electrical efficiency by 1.23%.
[83]	Active Cooling	PVT	Thermal efficiency increases by 1.96%. Panel efficiency increased by 1.5%.
[84]	Active Cooling	Thermoelectric radiant	Temperature reduced from 3–8 °C. Average electrical efficiency improvement is 2.6%.
[85]	Active Cooling	Nano-fluid	Temperature reduction 18.5 °C. Improvement in efficiency is 1.17%.
[86]	Passive Cooling	Water	Heating rate and cooling rate is operated experimentally. Average difference in temperature is 10 °C.
[87]	Active Cooling	Thermo-electric model	The average RMSE is 1.75 °C. The MAE is 1.14 °C. Experimental data is validated with MATLAB/Simulink.
[88]	Active Cooling	Water Heat Exchanger	Numerical results are compared with the experimental results. Reduction in average power output from the experiment is 4 W.
[89]	Passive cooling	PCM	Reduction in temperature is about 5 °C.
Present Study	Active and Passive Cooling	Fanless CPU Heat Pipe sink with Water	Average temperature achieved is 6.72 °C. Electrical efficiency had 2.98% improvement.

**Table 4 sensors-22-06367-t004:** Estimated cost of the experiment.

Items	Cooled PV (USD)	Uncooled PV (USD)
PV Panel	50	50
Aluminum Sheet	8	0
Fanless heat Pipe sink	14.25 × 4 = 57	0
Thermal grease	1	1
Silicone Gel	2	2
Thermocouples	2	2
**Total**	**120**	**55**

**Table 5 sensors-22-06367-t005:** Parameters used for LCE calculations.

Parameters	Cooled PV	Un-Cooled PV	Reference
Effective discount rate (i*_eff_*), %	5	5	[76]
Nominal escalation rate (r*_n_*), %	1	1	[76]
K*_O&M_*	0.96	0.96	Calculated
Capital recovery factor (CRF)	0.065	0.065	Calculated
Constant escalation levelized factor O&M, (CELF), %	0.9975	0.9975	Calculated
Annual operation and maintenance cost (C*_O&M_*), $	3.75	3.75	Calculated
Lifetime of the years (n), years	25	25	Assumed
Total investment cost (C*_inv_*), $	120	52	Calculated
Levelized cost of fuel (LC*_f_uel*), $/kWh	0	0	-

Note: The cost was taken into consideration at the time of the purchase of materials; it may increase or decrease from the present market price.

**Table 6 sensors-22-06367-t006:** Embodied energy for PV panels.

Component	Quantity	Energy Density (kWh/kg)	Cooled PV Panel (kWh)	Uncooled PV Panel (kWh)
PV Panel	0.4275 m^2^	999 kWh/m^2^	427.5	427.5
Aluminum Sheet	1.5 kg	4.11	6.165	-
CPU Pipe	0.600	8.1	4.86	-
Total Embodied Energy (kWh)	-	-	438.525	427.5
Annual Energy (kWh)	-	-	41.5735	35.5145
Energy Payback Time (Yr)			10.54	12.03

## Data Availability

All data used for the analysis are provided in the text.

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
