# Peer review of "Experimental Study on Performance Enhancement of a Photovoltaic Module Incorporated with CPU Heat Pipe—A 5E Analysis"

_sensors, 2022, doi:10.3390/s22176367_

Round 1
Reviewer 1 Report
I suggest revising the title and making it more concise. There is no need to describe every terminology in the title. If necessary, put important terms in the keywords. In my opinion, the title should be appealing.
What analysis is provided in 2.3.2, 2.3.3, and 2.3.4 other than writing equations? I believe there should be some analysis explaining the results related to the experiment.
In the introduction beginning, mention some emerging photovoltaic performances and include the suggested studies.
Indoor Organic Photovoltaics: Optimal Cell Design Principles with Synergistic Parasitic Resistance and Optical Modulation Effect, 2021, Adv. Energy Materials by Shim et al;
Recent developments in dye-sensitized photovoltaic cells under ambient illumination, 2021, Dyes and Pigments by Lee et al;
Energy recycling under ambient illumination for internet-of-things using metal/oxide/metal-based colorful organic photovoltaics, 2021, Nanotechnology by You et al;
Please improve the quality of the graphs. Their font size, legends, and axis need to be improved. All the graphs should be consistent.
How did the authors calculate the electrical efficiency of modules?
Language needs to be improved. There are many typos and grammatical errors. Also, no consistency in using hours, hr, minutes, min, subscripts, superscripts, etc.
The economic analysis presented in section 3.7 is not clear. Please shed a light on it.
Author Response
Response to Reviewer 1 Comments
Authors will want to express our profound gratitude to you and the respected Reviewer for the swift and in-depth nature of the reviewing process.
|
Reviewer comment |
Authors response |
|
Reviewer A |
|
|
|
|
|
1) 1) I suggest revising the title and making it more concise. There is no need to describe every terminology in the title. If necessary, put important terms in the keywords. In my opinion, the title should be appealing. |
Authors are very grateful to you for your in-depth assessment of the paper which helped us to improve our work. Finally, title is modified accordingly “Experimental Study on Performance Enhancement of a Photovoltaic Module Incorporated with CPU Heat Pipe- a 5E analysis” |
|
2) What analysis is provided in 2.3.2, 2.3.3, and 2.3.4 other than writing equations? I believe there should be some analysis explaining the results related to the experiment. |
Thanks for the comment. Those section as you mentioned present the mathematical relations which fall under the methodological section of the study, they are used for the analysis whose results are presented in the results and discussion section. The analysis of the exergy, energy, and energy payback time is presented in as given below in the revised version. a) Exergy analysis results section 3.5. (Fig.16). b) Energy payback time results section 3.8. They are therefore discussed appropriately. Thank You
|
|
3) In the introduction beginning, mention some emerging photovoltaic performances and include the suggested studies.
Indoor Organic Photovoltaics: Optimal Cell Design Principles with Synergistic Parasitic Resistance and Optical Modulation Effect, 2021, Adv. Energy Materials by Shim et al;
Recent developments in dye-sensitized photovoltaic cells under ambient illumination, 2021, Dyes and Pigments by Lee et al;
Energy recycling under ambient illumination for internet-of-things using metal/oxide/metal-based colorful organic photovoltaics, 2021, Nanotechnology by You et al |
Thanks for the suggestion. The reference mentioned by the reviewer are important and useful paper for the recent developments in solar photovoltaic module. Therefore, according to reviewer suggestion authors cited all three articles in the paper
For reference page 3 reference number 39 to 41 in the revised version.
Thank you.
|
|
3) Please improve the quality of the graphs. Their font size, legends, and axis need to be improved. All the graphs should be consistent. |
Authors agree with reviewer comment and we improved all figures as suggested, Thank you. |
|
4) How did the authors calculate the electrical efficiency of modules? |
Thanks for the comment. The electrical efficiency of PV modules calculated using Eq 11 in the revised paper. |
|
Language needs to be improved. There are many typos and grammatical errors. Also, no consistency in using hours, hr, minutes, min, subscripts, superscripts, etc. |
Authors agree with you, in revised we improved English and maintained consistency hours, hr, minutes, min, subscripts, superscripts in the article. |
|
The economic analysis presented in section 3.7 is not clear. Please shed a light on it. |
Thanks for the comment, the economic section is well discussed and explained in our humble. How the figures were arrived at were all backed with appropriate mathematical relations, the data used for the economic calculations are all provided in the tables 4 and 5. Some few lines are however added to make the sentences clearer. |

Reviewer 2 Report
I propose to make the following corrections to the work:
1. Please standardize the way of bibliographic entries, e.g.
line 38: is [1]-[3] should be [1-3], etc. or line 40: is [4], [5] and should be [4,5], etc.
2. Figures 4 and 5 have identical descriptions, but show different diagrams, and have illegible legends and axis descriptions in the wrong places.
3. Figure 10 incorrectly marks parts of graphs (a) and (b) for voltage and current graphs, respectively.
4. Appendix A is related to the main text and, due to its small scope (two charts), should not be singled out as a separate element of the work.
Author Response
Response to Reviewer 2 Comments
Authors will want to express our profound gratitude to you and the respected Reviewer for the swift and in-depth nature of the reviewing process.
|
Reviewer comment |
Authors response |
|
Reviewer B |
|
|
1) Please standardize the way of bibliographic entries, e.g.line 38: is [1]-[3] should be [1-3], etc. or line 40: is [4], [5] and should be [4,5], etc. |
Thanks for careful reading. Authors followed bibliographic entries accordingly to the reviewer suggestion |
|
2) Figures 4 and 5 have identical descriptions, but show different diagrams, and have illegible legends and axis descriptions in the wrong places. |
Authors would like to thank reviewer efforts and careful observation. In revised version authors showed completely different diagrams, legends, and axis are also rearranged.
|
|
3) Figure 10 incorrectly marks parts of graphs (a) and (b) for voltage and current graphs, respectively. |
Thanks for the comment. Please, the reviewer comment not clear to authors. Still, if authors understood clearly in revised version, fig.13 shows that a) is voltage and b) is current.
|
|
3) Appendix A is related to the main text and, due to its small scope (two charts), should not be singled out as a separate element of the work. |
Authors agree with reviewer comment, they are moved into the main text of the work. Thank you. |

Round 2
Reviewer 1 Report
I believe the article can be accepted now. It has been enough improved.